# Biopolymeric Innovations in Ophthalmic Surgery: Enhancing Devices and Drug Delivery Systems

**DOI:** 10.3390/polym16121717

**Published:** 2024-06-16

**Authors:** Kevin Y. Wu, Sameer Khan, Zhuoying Liao, Michael Marchand, Simon D. Tran

**Affiliations:** 1Department of Surgery, Division of Ophthalmology, University of Sherbrooke, Sherbrook, QC J1G 2E8, Canada; yang.wu@usherbrooke.ca (K.Y.W.); michael.marchand-gareau@usherbrooke.ca (M.M.); 2Department of Biology, Carleton University, Ottawa, ON K1S 5B6, Canada; 3Department of Biology, McMaster University, Hamilton, ON L8S 4L8, Canada; 4Faculty of Dental Medicine and Oral Health Sciences, McGill University, Montreal, QC H3A 1G1, Canada

**Keywords:** biopolymers, biocomposites, biomaterials, ophthalmology, materials science, oculoplastic surgery, orbital floor fracture, ocular prosthesis, hydrogels, ocular drug delivery

## Abstract

The interface between material science and ophthalmic medicine is witnessing significant advances with the introduction of biopolymers in medical device fabrication. This review discusses the impact of biopolymers on the development of ophthalmic devices, such as intraocular lenses, stents, and various prosthetics. Biopolymers are emerging as superior alternatives due to their biocompatibility, mechanical robustness, and biodegradability, presenting an advance over traditional materials with respect to patient comfort and environmental considerations. We explore the spectrum of biopolymers used in ophthalmic devices and evaluate their physical properties, compatibility with biological tissues, and clinical performances. Specific applications in oculoplastic and orbital surgeries, hydrogel applications in ocular therapeutics, and polymeric drug delivery systems for a range of ophthalmic conditions were reviewed. We also anticipate future directions and identify challenges in the field, advocating for a collaborative approach between material science and ophthalmic practice to foster innovative, patient-focused treatments. This synthesis aims to reinforce the potential of biopolymers to improve ophthalmic device technology and enhance clinical outcomes.

## 1. Introduction

The interface between material science and ophthalmology is currently undergoing a transformative phase with the innovative incorporation of biopolymers in the fabrication of medical devices. This review discusses the significant impact of biopolymers on the enhancement and development of critical ophthalmic devices, including intraocular lenses, stents, and prosthetics. Biopolymers possess exceptional properties, such as high biocompatibility, mechanical strength, and biodegradability, which make them superior alternatives to traditional materials.

This review examines the current utilization of biopolymers in ophthalmic device engineering. The unique physical properties, compatibility with biological tissues, and clinical performance of these biopolymers were assessed, along with a critical analysis of their roles and effectiveness. This review further delves into the specific applications of these biopolymers in oculoplastic and orbital surgeries, while also highlighting the innovative use of hydrogels in ocular therapeutics and discussing the emerging field of polymeric drug delivery systems designed for a variety of ophthalmic conditions.

## 2. Overview of Biopolymers and Biocomposites

### 2.1. Natural vs. Synthetic Biopolymers

Natural biopolymers are macromolecules derived from living organisms, including plants, animals, and microorganisms. They exhibit unique properties, such as biocompatibility, biodegradability, and low immunogenicity, making them ideal for biomedical applications. Examples include polysaccharides, proteins, and microbial polymers. These biopolymers can be used for drug delivery, tissue engineering, and as components of medical devices due to their inherent biochemical and biophysical properties such as cellular adhesion and degradation [1,2,3].

Synthetic biopolymers are engineered polymers created in a laboratory setting, often by modifying natural biopolymers or synthesizing new polymers from monomers. They are designed to mimic the properties of natural biopolymers and are used in various biomedical applications due to their stability, controlled release, and non-immunogenicity. These polymers can be tailored for specific purposes, such as drug delivery systems, implants, and tissue engineering, and are advantageous over natural biopolymers in terms of flexibility and stability for diverse applications [1,4,5].

### 2.2. Biocomposite Materials

Biocomposite materials are composed of natural fibers or particles embedded in a polymeric matrix. These materials are designed to mimic natural living tissues, thus offering superior biocompatibility, biodegradability, and non-toxicity. They are often used in medical devices, tissue engineering, and as carriers for bioactive molecules due to their structural integrity and functional mimicry of biological tissues. The combination of natural and synthetic elements in biocomposites allows for a balance between mechanical strength and biological compatibility, making them ideal for a variety of biomedical applications such as bone regeneration, orthopedic and dental implants, wound healing, and tissue engineering [6,7,8].

### 2.3. Nanoparticles-Based Polymeric Biomaterials

Nanoparticle-based polymeric biomaterials are nanoscale materials that combine the versatile properties of polymers with the unique features of nanoparticles. These biomaterials are widely used for targeted drug and gene delivery, tissue engineering, and bioimaging due to their enhanced biocompatibility, biodegradability, and ability to be precisely engineered. They can be designed to control the release rate of loaded molecules, optimize particle size and surface charge, and enhance cellular interactions. This makes them particularly effective in clinical applications, where precise targeting and controlled release of therapeutic agents are critical [9,10,11].

## 3. Oculoplastic and Orbital Surgery Devices

### 3.1. The Applications of Biomaterials in the Repair of Orbital Floor Fractures

#### 3.1.1. Overview of Orbital Floor Fractures and Ideal Properties of Biomaterials for Surgical Reconstruction

Orbital floor fractures, frequently resulting from impacts on the facial bones, are among the most common emergencies in trauma care, often leading to herniation and entrapment of orbital contents into the maxillary sinus. These fractures are typically caused by objects larger than the orbital aperture, with the hydraulic and buckling mechanisms explaining the resultant fractures [12,13,14,15,16]. Common symptoms include eyelid ecchymosis, edema, pain, restrictions in eye movement, and vertical diplopia. Large fractures may lead to enophthalmos, ptosis, and infraorbital hypoesthesia [12,17]. Orbital floor repair often employs implants ranging from alloplastic materials such as porous polyethylene, polytetrafluoroethylene, silicone sheets, or titanium mesh to autogenous options such as split cranial bone or iliac crest bone. The versatility of alloplastic implants is particularly beneficial for managing extensive fractures, although the optimal material choice continues to be debated [18].

The need for surgical intervention in orbital floor fractures depends on persistent symptoms and anatomical considerations. Indications for surgery typically include persistent diplopia with limitations in upgaze and/or downgaze, positive forced duction tests, enophthalmos exceeding 2 mm, and large fractures involving at least half of the orbital floor [12,17]. The surgical approach commonly involves an inferior transconjunctival incision, elevation of the periorbital region, liberation of herniated extraocular muscles and fat, and placement of an implant to prevent recurrent herniation. This method effectively addresses the functional entrapment of tissues, particularly impacting the inferior rectus muscle, and aims at aesthetic and functional restoration [18,19,20].

The primary goal of reconstructing the orbital wall is to re-establish normal anatomical relationships within the orbit. This task is complex and requires materials that not only fit the unique contours of the orbital cavity but also meet a spectrum of properties to ensure both safety and efficacy.

The ongoing search for an ideal biomaterial is steered by several key properties, including:
Biocompatibility and Non-Toxicity: The ideal biomaterial should not induce allergic reactions or have carcinogenic potential. Furthermore, it is essential that it closely mimics the physical properties of the native orbital tissue, thereby ensuring seamless integration without undue stress or strain on adjacent structures.Long-term Acceptance: The material must be capable of being permanently accepted by the body. This implies that it should not elicit a chronic inflammatory response or be subject to rejection.Chemical Stability: The selected material must be chemically inert. This stability is crucial, as the material needs to withstand sterilization processes without degradation of its chemical properties, thus ensuring long-term functionality and safety within the complex orbital environment.Manipulability and Stability: During surgery, the ease of manipulation of the material is vital for precise placement and shaping. Once implanted, it should maintain its form and integrity, and resist any deformation that could compromise the reconstructed anatomy.Fixation Capability: Effective fixation to the host bone is fundamental for long-term implant success. Therefore, the material should be amenable to secure fixation using various methods, such as screws, wires, sutures, or adhesives, to provide stability and prevent displacement.Anti-Microbial and Bone Preservation: It is important that the material does not promote microbial growth, which could lead to infection, nor should it promote resorption of the underlying bone. Additionally, it must not distort or exert undue pressure on adjacent structures, thereby maintaining the integrity of the orbital contents.Radiopacity: For effective postsurgical assessment, the material should be radiopaque, allowing for clear visualization during radiological evaluations. This feature is essential for monitoring the position and condition of the implant over time.Cost-Effectiveness: While ensuring high standards of quality and functionality, the materials should also be cost-effective. Balancing quality and cost is key to making these essential medical devices available to a broader patient population, thereby enhancing their overall well-being and quality of life.


#### 3.1.2. Current Biopolymers and Biocomposites for Orbital Floor Repair (Table 1) 

Absorbable Polymers:

Absorbable polymers have emerged as significant advancements in the repair of orbital fractures, offering a blend of biocompatibility and effective structural support. Among these, polylactic acid (PLA) and its derivatives have been extensively used because of their ability to provide mechanical integrity while gradually resorbing, thereby reducing donor-site morbidity and avoiding complications associated with permanent implants. For instance, Esmail et al.’s study on PLA (Resorb X) in orbital floor blow-out fractures revealed improved outcomes such as elevation, diplopia, and enophthalmos in most cases, although some patients experienced late enophthalmos and floor bowing after one year [21]. Similarly, polyglactin 910, or vicryl, is another prominent bioresorbable material commonly used in surgical sutures and has been adapted for orbital fracture repair with varying degrees of effectiveness [22,23].

However, the use of absorbable polymers for orbital reconstruction is limited. The resorption rates of these materials can vary, affecting their long-term stability and effectiveness. For example, PLA-based materials, while providing initial structural support, have shown significant floor bowing in medium-sized defects after one year, indicating potential limitations in their use [21]. Moreover, materials such as polydioxanone (PDO) have demonstrated variable outcomes, with several studies reporting complications such as hematoma, diplopia, and extrusion, especially in larger defects [24,25,26,27,28]. These limitations highlight the need for careful consideration of the material’s properties, defect size, and the specific requirements of orbital repair when choosing absorbable polymers for surgical applications. 

Non-Absorbable Polymers:

Non-absorbable polymers have long been used in orbital floor reconstruction because they are highly valued for their durability and mechanical properties. For instance, silicone has been a popular choice for nearly half a century due to its inertness and flexibility, with studies such as that of Prowse et al. highlighting its reduced postoperative complications and need for subsequent surgeries [29]. However, its use has declined due to complications such as cyst formation and extrusion [30,31,32].

Porous polyethylene (PE), particularly Medpor, has gained attention for its biocompatibility and strength. In a recent study by Marella et al., the effectiveness of titanium mesh versus Medpor implants in orbital floor reconstruction was compared. The study found that Medpor^®^ was more effective in reducing pain and enophthalmos than titanium mesh. However, for all other evaluated parameters, both groups showed comparable outcomes [33]. Despite its general acceptance, complications such as implant extrusion and infection have been reported [34,35,36].

Furthermore, the growing use of patient-specific 3D-printed models has further enhanced PE’s application. Pang et al. studied the use of Medpor, guided by a 3D-printed customized model, for the reconstruction of orbital floor fractures in two patient cases. These findings indicated a reduction in operative time and implant failure risk [37]. 

Recent advancements have led to the introduction of composite materials such as Medpor Titan, which combine the strength of titanium with PE’s tissue integration capabilities. Peng et al. found this hybrid material to be effective, with complication rates similar to those of titanium-only implants [38]. Additionally, HAPEX™, a composite of high-density PE and hydroxyapatite, has been recognized for its stiffness and osteoconductivity. However, there are limitations regarding its strength and brittleness in load-bearing bone substitute applications [39].

Despite the overall effectiveness of non-absorbable polymers, their usage is tempered by potential complications, highlighting the need for careful material selection and surgical techniques in orbital reconstruction.

**Table 1 polymers-16-01717-t001:** Summary of current biopolymers and biocomposites for orbital floor repair.

Material	Characteristics	Advantages	Limitations	References
Polylactic Acid (PLA)	- Biocompatible. - Gradually resorbing.	- Reduces donor-site morbidity. - Avoids complications of permanent implants.	- Late enophthalmos. - Floor bowing in medium-sized defects.	[21]
Polyglactin 910 (Vicryl)	- Commonly used in surgical sutures.	- Varying degrees of effectiveness in orbital fracture repair.	- Varying resorption rates affecting long-term stability.	[22,23]
Polydioxanone (PDO)	- Biocompatible. - Absorbable.	- Initial structural support.	- Hematoma. - Diplopia. - Extrusion in larger defects.	[24,25,26,27,28]
Silicone	- Inert. - Flexible.	- Reduced postoperative complications. - Fewer subsequent surgeries.	- Cyst formation. - Extrusion.	[29,30,31,32]
Porous Polyethylene (Medpor)	- Biocompatible. - Strong.	- Effective in reducing pain and enophthalmos. - Compatible with 3D printing.	- Implant extrusion. - Infection.	[33,34,35,36,37]
Medpor Titan	- Composite of titanium and PE.	- Combines strength of titanium with tissue integration capabilities of PE.	- Complication rates similar to titanium-only implants.	[38]
HAPEX™	- Composite of high-density PE and hydroxyapatite.	- Stiffness. - Osteoconductivity.	- Brittleness in load-bearing applications.	[39]

#### 3.1.3. Emerging Biopolymers, Biocomposites, and their Applications for Orbital Floor Reconstruction (Table 2)

Emerging biopolymers and biocomposites have shown remarkable potential for application in orbital floor repair, offering a blend of innovative solutions and enhanced biocompatibility. For instance, poly(trimethylene carbonate) (PTMC) has emerged as a significant material, particularly when combined with biphasic calcium phosphate particles and supplemented with titanium mesh. This combination is utilized in the design of patient-specific implants (PSIs) using 3D printing and offers enhanced customizability and precision in surgical applications. In a recent study, Guillaume et al. demonstrated that Osteo-PTMC implants expedited neovascularization and bone growth in the orbital area, marking a notable advancement over traditional titanium mesh. However, the long-term efficacy of such polymers requires further exploration to confirm their effectiveness in clinical settings [40].

Similarly, the use of bone marrow-derived mesenchymal stem cells (BMSCs) in conjunction with biomaterials such as β-TCP has shown promise in establishing osteoconductive environments for improved tissue regeneration and accelerated healing [41]. Wang et al.’s study on canine orbital defects using BMSC-BM/β-TCP showcased significant tissue regeneration, surpassing control groups in efficacy [41]. Despite these promising results, the challenge lies in translating these findings into clinical applications in humans.

Other emerging biomaterials involve the incorporation of nanoparticles, such as HA, into scaffolds for orbital floor repair. For instance, Alhamoudi et al. highlighted the integration of HA into polyurethane scaffolds, demonstrating enhanced mechanical strength and biocompatibility, particularly with 40% nano-HA incorporation [42]. Similarly, Sarfaraz et al. studied the use of Ce-doped ZnO nanoparticles in silk fibroin scaffolds. Their results indicated strong antibacterial properties and favorable biocompatibility [43]. However, the challenges associated with biocomposites include potential cytotoxicity and unclear stability, particularly in the context of antimicrobial metal ions. Another concern is the short half-lives of some nanoparticles, necessitating the development of controlled-release coatings to optimize their therapeutic potential. The emerging trend towards dual-action coatings that can adapt to the physiological environment promises to address some of these challenges but remains in the early stages of research [44].

Overall, while emerging biopolymers and biocomposites offer significant potential for orbital floor repair, their full potential can only be realized through further research and careful consideration of their long-term efficacy.

**Table 2 polymers-16-01717-t002:** Summary of emerging biopolymers and biocomposites for orbital floor repair.

Material	Characteristics	Advantages	Limitations	References
Poly(trimethylene carbonate) (PTMC)	Combined with biphasic calcium phosphate particles and titanium mesh	Enhanced customizability, precision, neovascularization, bone growth	Requires further exploration for long-term efficacy	[40]
Bone marrow-derived mesenchymal stem cells (BMSCs) with β-TCP	Establishes osteoconductive environments for tissue regeneration	Improved tissue regeneration, accelerated healing	Challenge in translating findings to human clinical applications	[41]
HA nanoparticles in polyurethane scaffolds	Incorporation of nanoparticles	Enhanced mechanical strength, biocompatibility	Potential cytotoxicity, unclear stability, short half-life	[42]
Ce-doped ZnO nanoparticles in silk fibroin scaffolds	Nanoparticles with antibacterial properties	Strong antibacterial properties, favorable biocompatibility	Potential cytotoxicity, unclear stability	[43]
Dual-action coatings	Adaptable to physiological environment	Promising to address current challenges	Early stages of research	[44]

### 3.2. The Applications of Biomaterials in Ocular Prosthesis

#### 3.2.1. Overview of Ocular Prosthesis and Ideal Properties of Biomaterials for Restoring Functionality and Aesthetics

Ocular prostheses are essential for restoring facial symmetry and aesthetics in patients who have undergone ocular surgeries such as evisceration, enucleation, or exenteration. These prosthetic devices are essential for cosmetic rehabilitation, particularly following enucleation, where the entire eye is removed to treat conditions such as retinoblastoma and choroidal melanoma. Evisceration, which involves removing the globe’s contents but preserving other ocular structures, and exenteration, the more extensive re-movement of orbital tissues, also necessitates ocular prostheses for aesthetic restoration [45,46]. These prostheses are custom-fitted 4–12 weeks post-operation and are essential for creating a natural appearance, matching the unoperated eye in terms of color, size, and position, thereby significantly enhancing the patient’s quality of life.

The choice of material for ocular prostheses is a critical aspect of their functionality and aesthetic value. Inert materials, such as glass, silicone, or methylmethacrylate, are often used owing to their comfort, low extrusion rates, and cost-effectiveness. These materials are suitable for use in patients who do not require implant integration. However, advances in biomaterials have led to the development of biointegrated materials such as hydroxyapatite, porous polyethylene, and aluminum oxide. These materials are designed to integrate into the soft tissue of the socket, allowing for the direct attachment of extraocular muscles. This integration is important for achieving natural eye movements and enhancing cosmetic outcomes, although it may increase the risk of postoperative complications such as exposure and infection [47]. The selection of these materials is based on individual patient needs, with a focus on achieving the best possible functional and aesthetic results.

An ideal ocular prosthesis constructed with biomaterials must possess several critical properties to ensure its suitability and efficiency. This is essential for an ocular prosthesis to be a viable and practical solution for those requiring it. Some of these properties include the following:Lightweight and Comfort: The prosthesis should be designed to be lightweight to promote maximum comfort for the patient. A heavy prosthesis can lead to discomfort and may cause strain on the surrounding orbital tissues. Therefore, the use of lightweight materials ensures that patients can perform their daily activities without feeling discomfort. It is important that this lightness in weight is achieved without compromising the durability or functionality of the prosthesis. An ideal lightweight prosthesis should be comfortable for prolonged wear while maintaining its structural integrity and functionality.Color Match to the Contralateral Eye: The prosthesis should be custom-tailored in color to match the characteristics of the contralateral eye as closely as possible. This approach ensures that the artificial eye is virtually indistinguishable from the natural eye color, thereby enhancing the overall aesthetic appearance.Texture and Integration with Facial Features: The prosthesis should mimic the natural eye not just in color but also in texture. This means that the surface of the prosthesis should feel similar to that of the natural eye when touched. This attention to detail in replicating the natural eye texture contributes significantly to a natural appearance and feel, enhancing the prosthesis’s integration with natural facial features.Hygiene and Maintenance: The design of the prosthesis should allow for easy and effective cleaning. Good hygiene is important to prevent infections and maintain the health of the surrounding orbital tissues. The surface and material of the prosthesis should not hold onto the bacteria and should be resistant to the build-up of deposits.Availability and Accessibility: The prosthesis should be readily available for re-placements or adjustments as needed. This means that the manufacturing processes should be sufficiently efficient to ensure that these prostheses are easily accessible to those in need. Availability is key to ensuring that patients can quickly obtain replacement or adjustment if their prosthesis becomes damaged or their physical needs change. This accessibility is essential for the continuous and comfortable use of prostheses, ensuring that patients do not face long periods of discomfort or lack of functionality.

#### 3.2.2. Current Biopolymers and Biocomposites in Ocular Prosthetics

Poly(methylmethacrylate) (PMMA), a prevalent polymer in ophthalmic applications, is highly valued for its biocompatibility and visibility in ocular prostheses [48]. While PMMA dominates the market for both off-the-shelf and custom-made prosthetic eyes due to its durability, longevity, and tissue compatibility [49], its inherent hydrophobicity can lead to challenges such as poor wettability and tear protein deposits, causing patient discomfort and anophthalmic socket conditions [50]. To address this, Pine et al. investigated derivatives of poly(ethylene glycol) (PEG) and methyl methacrylate (MMA) grafted onto PMMA. Their findings indicated that ethylene glycol dimethylacrylate (EGDMA)-grafted PMMA significantly improved hydrophilicity, enhancing wettability by 33% compared to polished PMMA prostheses, thus potentially mitigating PMMA’s hydrophobic limitations in ocular prosthetics [51].

Alphasphere is an FDA-approved orbital implant made of poly-HEMA (2-hydroxyethyl methacrylate). Its unique two-phase structure, which combines a porous sponge front with a nonporous gel back, enhances prosthesis placement and maneuverability while mitigating tissue integration risks [52]. In a study, Shevchenko et al. demonstrated its effectiveness in managing anophthalmic sockets, offering benefits such as direct muscle suturing and reduced exposure risks. However, a few cases have indicated potential complications associated with Alphasphere. For instance, Yadav et al. presented a case in which a patient with diabetes experienced a significant and irreversible reaction to an Alphasphere implant. This reaction occurred without any signs of infection, suggesting that the implant itself might have triggered an adverse response in the host tissue [53]. Furthermore, Neimkin et al. documented a case in which a patient underwent enucleation for choroidal melanoma and received an Alphasphere implant. The patient initially recovered without issues, but after two years, they experienced implant failure [54]. These potential complications highlight the need for further research and monitoring of the long-term efficacy and safety of Alphasphere implants in ocular prosthesis applications.

#### 3.2.3. Emerging Biopolymers, Biocomposites, and Their Applications in Ocular Prosthetics

Emerging biomaterials and novel techniques have shown remarkable potential for application in ocular prosthetics. These include surface coatings to enhance antimicrobial activity in ocular prostheses.

Preventing bacterial colonization on ocular prostheses is a crucial aspect of postsurgical care, as it significantly reduces the risk of infections that could lead to additional surgical interventions. This has led to growing interest in the development of surface coatings designed to enhance the antimicrobial properties of these implants and prostheses. One innovative approach involves the integration of Ag and Cu ions and nanoparticles into orbital implants. A noteworthy example is seen in a study conducted by Yang et al., where ocular prostheses made from PMMA resin incorporated with silver concentrations ranging from 300 to 700 ppm displayed remarkable antibacterial properties. This resulted in an impressive 99.9% reduction in bacterial growth, highlighting the potential of Ag for this application [55].

In another study, Baino et al. explored the use of a silver nanocluster/silica biocomposite in a PMMA prosthesis. This biocomposite not only exhibited strong adhesion qualities but also demonstrated potent antibacterial activity against Staphylococcus aureus. The primary advantage of this approach is the use of metal ions instead of conventional antibiotics, which reduces the likelihood of bacterial resistance and toxicity. The release of silver ions, as opposed to nanoparticles, is a key characteristic of this approach [56]. Ye et al. introduced a CuO-doped mesoporous bioactive glass (Cu-MBG) biopolymer coating on HA implants, which combined anti-inflammatory and antimicrobial effects through its expansive drug-loading capacity [57].

These advances in surface coatings not only improve the efficacy of orbital implants but also allow for better prevention of postoperative complications, specifically infections resulting from bacterial colonization of ocular prostheses.

### 3.3. The Applications of Biomaterials in Posterior Lamellar Eyelid Reconstruction

#### 3.3.1. Overview of Posterior Lamellar Eyelid Reconstruction and Ideal Properties for Tarsal Substitutes

The posterior lamella of the eyelid, which comprises the palpebral conjunctiva and tarsal plate, plays a pivotal role in maintaining ocular integrity and ensuring corneal protection. The tarsal plate, characterized by its cartilage-like connective tissue, provides the eyelid with mechanical strength due to its rich extracellular matrix composed of collagen and glycosaminoglycans [58,59,60,61,62]. Additionally, the palpebral conjunctiva, with its stratified epithelium and vascularized basement membrane, contains goblet cells essential for tear film maintenance [63,64,65]. Integral to this structure are the meibomian glands embedded in the tarsal plate, secreting meibum crucial for tear film composition and ocular comfort [66,67]. The intricate anatomy of the marginal eyelid, including features such as the Riolan’s muscle and accessory glands of Zeis and Moll, highlights the complexity and vital functionality of the eyelid in ocular health (Figure 1).

In terms of reconstruction, the approach is primarily determined by the extent of the eyelid defect. Defects of less than 33% of the eyelid’s total width are generally managed through primary closure, possibly augmented with lateral canthotomy or superior cantholysis. However, larger defects exceeding this threshold necessitate more involved reconstruction techniques, including the use of grafts and flaps [69,70,71,72]. The key objectives of such procedures are to preserve eyelid function, ensure stable margins, and provide adequate closure for ocular protection while also considering aesthetic outcomes. This involves careful matching of the donor tissue to the defect, particularly in the anterior lamellar region, adhering to the “like for like” principle [69]. The complexity of reconstruction is amplified in the posterior lamellar region due to its distinct anatomical and functional characteristics, coupled with the limited availability of suitable donor tissues [73,74,75].

The ideal properties of a tarsal substitute include:Structural Integrity and Durability: A tarsal substitute should be thin enough to not cause any discomfort or disruption in the eye’s anatomy, yet stable enough to maintain its shape and function over time. This durability is crucial to ensuring that the substitute can withstand the mechanical forces exerted during blinking and eye movements without deformation or deterioration.Biocompatibility: High biocompatibility is essential for minimizing adverse reactions from the body’s immune system. A biocompatible tarsal substitute would reduce the risk of rejection and other complications, such as irritation or infection, ensuring safer integration with the surrounding tissues.Tissue Integration: The ability to seamlessly merge into the peripheral tarsus is vital for a successful implant. This integration ensures that the substitute behaves as a natural component of the eye, thereby facilitating normal eyelid function. They should bond well with surrounding tissues without causing any structural weaknesses or abnormalities.Anti-Inflammatory: The ideal tarsal substitute should not provoke any inflammatory response. Therefore, it must be designed to avoid triggering the body’s immune response, which can lead to swelling, redness, and discomfort, thereby ensuring a more comfortable and effective healing process.Biomimetic Functionality: Mimicking the physical structures and biological functions of the native extracellular matrix is important as a substitute for effective function. This involves replicating the texture, elasticity, and strength of the natural tarsal plate, as well as its ability to interact with native cells and tissues to promote normal eyelid function.Cellular Support: The substitute should foster cell survival, proliferation, and growth. This provides a conducive environment for cells to adhere, grow, and function normally. This should encourage healthy tissue regeneration and integration, thereby contributing to the overall success and longevity of implants.

#### 3.3.2. Current Types of Biomaterials and Approaches for Posterior Lamellar Eyelid Reconstruction

Various autogenous tissue grafts and flaps have been used for posterior lamellar eyelid reconstruction. For moderate lower eyelid defects, semicircular or adjacent tarsoconjunctival flaps combined with full-thickness skin grafts are common [76,77,78]. Larger defects often necessitate eyelid-sharing techniques such as the Cutler–Beard and modified Hughes flaps, which require a second surgery for eyelid reopening [79,80,81]. Free tarsoconjunctival grafts from the contralateral eyelid, covered with a skin–muscle flap, are also used, avoiding the need for a second surgery. However, some complications include ectropion and entropion and are unsuitable for those lacking sufficient redundant upper eyelid skin [82,83,84,85].

For grafts beyond the posterior lamella, alternative sites include the lip, buccal mucosa, gingival alveolar mucosa, hard palate, auricular cartilage, and nasal septum [86]. However, these alternative graft sources have unique characteristics and limitations. Oral mucosa grafts are distinguished by their high vascularization, which aids in graft integration but lacks goblet cells, leading to potential corneal irritation [87,88,89]. Hard palate mucoperiosteal grafts offer structural support similar to the posterior lamella, but may cause discomfort post-transplantation [90,91]. Nasal mucosal grafts, histologically similar to tarsoconjunctiva, provide a stable eyelid margin but may cause irritation due to nasal hair [91,92,93]. Auricular cartilage grafts, known for their durability, may cause corneal irritation due to their texture and lack of secretory function [94,95]. Despite the availability of these non-tarsoconjunctival autografts, various challenges exist in achieving functional matching, managing limited donor areas, and minimizing donor-site morbidity [96].

The selection of materials and methods for posterior lamellar eyelid reconstruction is determined by the size of the defect, availability of donor tissue, and specific needs of the eyelid region. Autogenous tissue grafts, such as tarsoconjunctival flaps and various mucosal and cartilage grafts, are standard materials used for reconstruction because of their compatibility and reduced risk of rejection. However, each graft type presents its own set of difficulties, including donor site morbidity, potential postoperative complications such as ectropion and entropion, and limitations in providing the necessary structural and functional properties for successful reconstruction. Future advancements in biomaterials and surgical techniques are expected to address these challenges with the aim of improving patient outcomes by enhancing graft integration, minimizing complications, and achieving more natural and functional eyelid reconstruction.

#### 3.3.3. Emerging Biopolymers, Biocomposites, and Their Applications for Posterior Lamellar Eyelid Reconstruction

In the search for more biocompatible and effective biomaterials for posterior lamellar reconstruction, several biomaterials have been identified as promising alternatives to traditional autografts. These materials have been explored for their unique properties and potential applications.

Natural Polymers:

Natural polymers, such as collagen (Col I) and chitosan, have shown promising results owing to their biocompatibility and adaptable properties. Collagen, prevalent in the conjunctival matrix, has been effective in conjunctival repair, as seen in rabbit models, promoting rapid healing with minimal scarring [97,98]. Drechsler et al. used compressed collagen for human conjunctival cell expansion, akin to amniotic membrane effects [99]. In another study, Sun et al. developed novel chitosan scaffolds for eyelid tarsus tissue engineering. The results indicated that the chitosan scaffolds provided support for the attachment and proliferation of NIH 3T3 mouse fibroblasts, as well as human orbital skin fibroblasts in vitro [100].

Furthermore, Xu et al. investigated a branched polyethylene (B-PE) elastomer for eyelid reconstruction, focusing on its biocompatibility and effectiveness. The B-PE scaffolds tested in vitro and in vivo showed no significant cytotoxicity, a mild inflammatory response, and promoted collagen deposition and fibrovascularization. B-PE also demonstrated comparable performance to traditional materials when used for eyelid reconstruction in rabbits, with fewer complications and better integration with surrounding tissues [101]. Despite these advances, challenges such as vascularization and nerve integration in grafts limit the clinical application of these materials.

Synthetic Polymers:

Synthetic polymers, due to their customizable properties, have gained attention for their potential applications in conjunctival reconstruction. For instance, Bosworth et al. utilized PCL in electrospun scaffolds combined with decellularized matrices for conjunctival defect repair and observed improved cellular layering [102]. Similarly, Yao et al. developed a composite scaffold using collagen and poly(L-lactic acid-co-ε-caprolactone) (PLCL), which showed strong suitability for treating conjunctival epithelial colobomas in vitro [103]. Yan et al. enhanced a poly(lactic acid) (PLA) scaffold with cellulose, silk peptide, and levofloxacin to achieve effective conjunctival repair in rabbit models [104]. Despite the potential of synthetic polymers for conjunctival reconstruction, some limitations include their inability to closely replicate the complex cellular microenvironment of natural tissues, which often results in suboptimal tissue integration and inconsistent cell growth.

In addition, synthetic polymers have been investigated for the reconstruction of tarsal plates. For example, high-density porous polyethylene has been used as an eyelid spacer in clinical settings. However, its application has some limitations, including postoperative complications such as issues related to implant stability, visibility under the skin, irregular contouring of the eyelid, and discomfort for the patient [105]. These challenges highlight the need for further research and careful consideration of material properties in the selection and design of synthetic implants for eyelid reconstruction.

The potential use of novel biopolymers and biocomposites in posterior lamellar eyelid reconstruction represents a significant advancement in the field and a promising alternative to conventional methods. Natural polymers, such as collagen and chitosan, exhibit great potential for healing and tissue integration, whereas synthetic polymers offer the advantage of customizable properties to meet specific reconstruction needs. However, challenges remain in achieving optimal tissue integration and managing postoperative complications. Ongoing research is necessary to further refine these materials and ensure their clinical effectiveness and safety in eyelid reconstruction.

## 4. Applications of Hydrogels in Ophthalmology

A hydrogel is a three-dimensional network of physically or chemically cross-linked hydrophilic polymers, giving it a unique capability to absorb and retain water. Hydrogels can be derived from biomaterials, synthetic materials, or hybrid materials. Their versatile compositions and functional design offer them various properties and advantages that can be applied across fields such as agriculture, the food industry, environmental engineering, and biomedicine. Specifically, biopolymer-based hydrogels are built from natural, biodegradable biopolymers, including proteins (fibrin, fibroin, collagen, keratin, and gelatin), polysaccharides (e.g., hyaluronic acid, cellulose, alginate, chitosan), and nucleic acids [106]. The unique nature of biopolymer-based hydrogels, such as their biocompatibility, biodegradability, non-toxicity, water-absorbing and retaining properties, and soft consistency, has made them a valuable tool in the biomedical field. Recent advancements have expanded their ophthalmic applications to include intraocular lenses, cell therapy, wound repair, and vitreous substitutes [106,107].

### 4.1. Intraocular Lenses (IOLs)

Intraocular lenses (or IOLs) are artificial lens implants designed to replace one’s natural lens during cataract surgeries. The primary purpose of an IOL is to replace the diseased or damaged natural IOL and restore clear vision. An ideal IOL should exhibit biocompatibility, optical clarity, and stability within the eye, with minimal post-operative complications [108,109]. 

Conventional IOL materials in the market are acrylic and silicone, delivering decent optical performance but face practical challenges. For instance, acrylic materials such as polymethylmethacrylate (PMMA) were favored for their low cost, optic clarity, and biocompatibility. However, PMMA lenses are non-flexible, cannot withstand high temperature and pressure, and the implantation requires a large incision (5–7 mm), leading to a high occurrence of posterior capsular opacification (PCO) [110,111]. Other materials for IOL, such as hydrophobic acrylic and hydrophilic acrylic, have later emerged with better flexibility and can be inserted with minor cuts around 2 mm. Hydrophobic acrylic-based IOLs are superior in optical quality and customizability. In terms of biocompatibility, they are associated with a low rate of PCO but a high rate of inflammatory cell accumulation. On the other hand, hydrophilic acrylic-based IOLs are associated with a low rate of inflammatory cell outgrowth but high rates of PCO and calcification and limitations in customizability [111,112]. Silicone IOLs are soft/flexible lenses capable of small incisions and biocompatibility of low inflammatory epithelial cell outgrowth on the lens. Silicone material has been less commonly used recently due to its drawbacks of causing anterior capsular fibrosis, opacification, and uncontrollable fast unfolding after insertion [111,113]. 

Addressing these challenges is crucial for enhancing patient comfort and optimizing treatment outcomes. Recent advancements have led to the development of state-of-the-art biopolymer-based hydrogel IOLs. Liu et al. took advantage of the self-polymerization property of dopamine (DA), modified the IOL surface with drug-eluting coating, and successfully loaded drugs such as the antiproliferative drug doxorubicin (DOX) and hydrophilic 2-methacryloyloxyethyl phosphorylcholine (MPC). The PolyDA(DOX)-MPC surface-coated drug-eluting IOLs were tested in vitro in rabbit eye models and in vivo. Results were reported to effectively prevent the formation of PCO by reducing surface cell and protein adhesion on the IOL. In addition, the biopolymer-coated IOLs were tested to be non-toxic and triggered no change in ocular structures or morphologies post-implantation [114]. Recent advancements in cellulose-based hydrogels have shown significant promise for their use in IOLs, particularly in preventing PCO. One notable study reported the development of a state-of-the-art modification of cellulose-based hydrogel IOLs that effectively prevents PCO. Yang et al., and their team synthesized the hydrogel using repeated freeze–thaw cycles, which enhanced its mechanical properties and transparency. Also, the hydrogel was modified with zwitterionic compounds to form thick hydration layers. The additional layers provide an antifouling effect, thereby inhibiting the adhesion and proliferation of epithelial cells that contribute to PCO [115].

Even though mucin is a natural biopolymer present in the eye, limited studies focus on its use in the fabrication of IOLs. Mucin contributes to lubrication, wettability, and serves as a protective barrier, playing a vital role in maintaining a healthy ocular surface. Additionally, mucin in the tear film traps debris and pathogens from the environment. Current ongoing research suggests that mucin-coating in the fabrication of contact lenses provides several benefits. These include reducing tribological damage to corneal tissues and irritation, providing a hydrophobic surface to reduce lipid absorption, and improving surface wettability while maintaining critical properties such as biocompatibility, transparency, and gas permeability [116,117]. Thus, mucin or mucin-like glycopolymer coatings might also provide benefits for intraocular lenses. However, more research is required to explore how mucins can enhance IOL function.

### 4.2. Vitreous Substitutes

The vitreous humor is a gel-like substance situated between the lens and the retina that occupies two-thirds of the eyeball volume. Vitrectomy and vitreous substitutes may be required in situations of vitreous liquefaction, severe eye trauma, infection, or vitreous hemorrhage. Traditional substitutes include gases (sulfur hexafluoride, perfluoropropane) and lipids (silicone oils, perfluorocarbon liquids, and balanced salt solutions). Though effective to some extent, it often falls short of mimicking the complex properties of the natural vitreous, such as transparency, permeability, viscosity, and the ability to maintain intraocular pressure and support the retina and other ocular tissues [118,119]. The introduction of biopolymer-based hydrogels offers the potential to replicate the unique characteristics of the vitreous humor more closely. Biopolymer-based hydrogels can be made from or composites of hyaluronic acids (HA), collagen, gelatin, chitosan, alginate, silk fibroin, etc. [118,119]. 

For instance, Baker’s team has engineered a vitreous substitute by cross-linking aldehyde or ketone-modified hyaluronan, a component of the natural vitreous, and poly-tetraoxyamine to better mimic the density, transparency, and refractive index of the natural vitreous. The team evaluated the ratio of aldehyde-HA to ketone-HA and oxyamine. The resulting HA-oxime hydrogel is demonstrated in a rabbit eye model to have a natural vitreous-matched transparency, long-term stability, refractive index, and is capable of easy injection and maintaining intraocular pressure with no signs of adverse side effects such as redness, cloudiness, and inflammation [120]. Another hyaluronan composite hydrogel, hyaluronan and methylcellulose (HAMC)-based hydrogel, can be a potential material for vitreous substitution due to their biocompatibility and non-toxicity with the eye. Recent research has employed HAMC hydrogels to deliver the drug intravitreally to the retina and effectively attenuate intraocular inflammation [121,122]. In addition, silk-hyaluronic acid composite hydrogels have been suggested as another potential vitreous substitute due to their matching refractive index, offering optical clarity and the ability to maintain intraocular pressure and prevent retinal detachment. Recent studies have extensively evaluated the effect of the silk-to-hyaluronic acid ratio on hydrogel swelling capability and stability. Nevertheless, more studies regarding its long-term stability, retinal cell cytocompatibility, and animal trials are required [123]. Other than HA-based hydrogel, alginate-based hydrogel has also been proposed as a vitreous substitute. Choi et al. engineered a transparent alginate phenylboronic acid/polyvinyl alcohol composite hydrogel (TALPPH) and evaluated its properties in vivo with rabbit eye models. Results demonstrated a matched refractive index with natural vitreous and a significantly superior refractive index to traditional silicone oil-based vitreous substitutes. Although it maintained a relatively consistent transparency of around 87% over time, it possesses a slightly lower transparency compared to natural vitreous and traditional vitreous substitutes. Additionally, the TALPPH hydrogel was effective functionally as a vitreous substitute in preventing retinal detachment and maintaining intraocular pressure without adverse effects such as post-operative cataracts and opacification [124]. Other compositions of alginate-hyaluronic acid-based hydrogels were also evaluated as potential vitreous replacements, where results showed high biocompatibility, high optical quality of >90% transparency, and similar viscoelastic properties as natural vitreous bodies [125].

Despite the advancement in applying biopolymer-based hydrogel as a potential vitreous substitute, many are still in animal and in vitro studies. Ongoing research is needed to continue unravelling the best formulation and ratio of biopolymer compositions in hydrogels to best resemble natural vitreous in humans. Clinical trials and studies of its potential immunogenic reactions such as inflammation, retinal adhesion, long-term residence stability, fast degradation problems, production techniques and standardization, and the potential for personalization of hydrogel formulations, are all areas of future research directions [118,119].

### 4.3. Ocular Wound Repair for Cornea Damage

Due to the shortage of replacement donor tissues and the unique transparent properties of optical tissues, such as the cornea, ocular wound repair/regeneration has been challenging for scientists. In the past decade, biopolymer-based hydrogels have been explored for their application in ocular wound healing due to their unique biocompatibility, biodegradability, designability as scaffolds, and bio-adhesive traits. This section dives into their role in repairing ocular damages, such as corneal regeneration, tissue engineering, and the broader paradigm of cell-based therapy.

Gelatin methacrylate (GelMA) and its modifiers have been extensively studied in ocular damage repair, specifically in corneal stromal regeneration. GelMA possesses excellent biocompatibility due to its fabrication from biomaterials while still maintaining good mechanical stability compared to synthetic material-based hydrogels. Its unique cell adhesive with RGD (arginine, glycine, and aspartate) patterns and MMP-degradable amino acid chain makes it useful in corneal stroma regeneration [126,127]. Kong et al. prepared fiber-reinforced GelMA hydrogel constructs to achieve similar properties in light transmittance, fiber spacing ratio, and swelling ratio compared to native corneal stroma structures. The team also demonstrated GelMA hydrogel’s potential in inducing corneal stromal matrix synthesis and regeneration in vitro and in vivo, in rat corneas [128]. Another recent study in 2022 investigated the use of GelMA hydrogel in sutureless keratoplasty for corneal damage. The team developed a photocurable bioadhesive hydrogel composed of GelMA and oxidized dextran; the hydrogel was evaluated subsequently to possess superior adhesive strength than conventional adhesives, high transparency for light to transmit, stability and resistance to enzymic degradation, and low rates of inducing inflammation [129]. Another composite hydrogel, formed by integrating GelMA with Pluronic F127 diacrylate, aldehyded Pluronic F127 micelles, and collagen type I, has demonstrated superior performance in fostering the generation of epithelium and stroma. This hydrogel exhibits noteworthy attributes, including high transparency, bio-adhesiveness, toughness, and prolonged degeneration time. In vivo, experiments in rabbit models showed evidence of GELMA-F127DA&AF127 hydrogel successfully integrating with the corneal tissues without the need for sutures, providing beds for corneal epithelium and stroma regeneration [130]. A derivative of GelMA, Gel for corneal regeneration (GelCORE), has been investigated extensively in the past few years due to its biocompatibility, bio-adhesiveness, transparency, and cost-effectiveness [131]. Khalil’s team expanded the potential of GelCORE in loading drugs such as the antibiotic ciprofloxacin and preventing eye infections and inflammation, in addition to their corneal regeneration role. The study developed GelCORE with ciprofloxacin-loaded micelles. In in vitro and ex vivo pig corneal injury models, drug-loaded GelCORE demonstrated effective drug-releasing capacity, leading to higher cell viability while not affecting the GelCORE’s original adhesiveness and compatibility [132]. This also showed the potential application of biopolymer-based hydrogel in drug delivery systems, which will be discussed in the next section (Section 5) of this review.

Collagen-based hydrogel has also been explored for tissue engineering and corneal regeneration. Collagen is a component found naturally in the human stroma; thus, collagen-based hydrogel exhibits excellent biocompatibility and could mimic similar structural properties with corneal tissues. Various research has been carried out regarding the use of collagen, specifically type I collagen, for corneal tissue engineering, as it is one of the most abundant types present in human connective tissues and the cornea [133]. For example, type I collagen–gelatin cross-linked hydrogel may be used in corneal regenerations and as an artificial corneal substitute in corneal tissue injury or other keratopathies [134]. Experimental results demonstrated that stem cell viability is higher in collagen–gelatin-based hydrogel compared to gelatin-based hydrogel while still maintaining a matched optical clarity, biocompatibility, and chemical and mechanical properties with natural cornea [135]. Another recent study developed a crosslinker-free collagen-based hydrogel for corneal regeneration using a self-assembly process involving a small peptide (PyKC), which effectively entrapped collagen without modification. This formulation resulted in a non-toxic, water-insoluble hydrogel that remained stable for over a year and was capable of trapping and protecting protein molecules from external denaturing agents. The study highlights its potential as an artificial cornea or as a protective sealant for corneal repair. that can remain insoluble for a prolonged period of time [136].

In particular, a recent study improved a drug-loaded collagen-based hydrogel by incorporating nanocellulose and employing both chemical and photochemical cross-linking to reinforce the structure. The modified hydrogel exhibited sustained anti-inflammatory drug release and was successful in reducing inflammation in a corneal disease model while maintaining corneal transparency. This enhancement utilized cellulose, an abundant biopolymer, integrated into another biopolymer-based hydrogel, ensuring its biocompatibility and sustainability [137]. Additionally, another study proposed the use of nanocellulose-reinforced poly (vinyl alcohol) (PVA) hydrogel as potential therapeutic contact lenses to facilitate corneal regeneration. Ex vivo experiments confirmed its biocompatibility; however, further studies are necessary to evaluate its long-term efficacy and safety before it can be used commercially or in human trials [138]. Thus, the integration of nanocellulose into hydrogel systems represents a promising avenue for biomaterial developments and corneal regeneration treatments.

Diverse biomaterials are utilized to fabricate hydrogels and have been tested for their potential for corneal regeneration. For instance, silk fibrin and polyacrylamide interpenetrating hydrogel could be employed for corneal stromal regeneration by facilitating keratocyte migration, proliferation, and up-regulation of keratocyte genes that are normally associated with healthy corneal stromal cells [139]. Advantages of silk fibrin include biocompatibility, rapid gelation, tunable degradable rate, silk protein transparency, and the presence of interconnective pores. In the murine model of corneal injury, Bhattacharjee’s team reported that TPCA-1-releasing silk fibroin hydrogel promoted corneal epithelium and stromal regeneration, where TPCA-1 is a selective inhibitor involved in the proinflammatory signaling NF-κB pathway. On the other hand, peptide-based hydrogels have also been widely studied for corneal repair due to their ability to mimic native ECM. For instance, a poly-ε-lysine (pεK)-based hydrogel may be cross-linked with a library of bis-carboxylic acid or octanedioic acid to achieve desirable properties such as porosity, transparency, and swell-capability [140,141]. In addition, pεK hydrogel has sufficient free amines to allow the covalent binding of EMC and synthetic cell binding peptides. A recent study by Kennedy et al. demonstrated the potential of using octanedioic acid cross-linked pεK hydrogels as a substrate for corneal endothelial cell growth [142]. 

There are studies developing mucin hydrogels for tissue engineering and drug delivery. However, no studies have yet to explore their ocular application. This gap in research highlights a significant opportunity, given the promising properties of mucin, such as its abundance, biocompatibility, and natural existence in the eye. Recent advancements demonstrate the potential of mucin in hydrogel formulations that could be beneficial for ocular therapies, particularly corneal regeneration, and repair. For example, one study showed that mucin hydrogels can enhance transparent viability through efficient cell encapsulation [143]. Another study created a pectin- and mucin-modified cellulose hydrogel for controlled curcumin release, offering tunable pH-sensitive drug release properties. However, this formulation lacks the transparency needed for ocular application [144]. A third study confirmed the biocompatibility of chitosan-co-mucin composite hydrogel and its potential for broader tissue engineering uses, including corneal regeneration [145].

The versatile composition and tunability of biopolymer-based hydrogels shed light on customizable hydrogels to meet specific patient needs. Despite their promising properties and broad application potential, many studies are carried out in cell cultures and animal models; research in biopolymer-based hydrogels is still ongoing to tackle challenges such as optimization and formulation of hydrogel composites, permeability, refining porosity, and controlling degradation time.

Table 3 summarizes recent applications of biopolymer-based hydrogels in corneal repair.

## 5. Biopolymeric Drugs Delivery Systems

### 5.1. Anatomical and Physiological Barriers in Ocular Drug Delivery

The administration of ophthalmic drugs is a multifaceted process that employs various methodologies, including but not limited to topical, systemic, periocular (i.e., subconjunctival), intravitreous, and suprachoroidal routes (Figure 2). These techniques are pivotal in managing numerous ocular diseases commonly encountered in clinical practice. Among these, topical application stands out as the most straightforward method, utilizing different forms like solutions, suspensions, ointments, gels, or emulsions. Despite its simplicity, the effectiveness of topical application is limited; approximately 5% of the administered dose successfully penetrates the eye’s internal structures [146].

The primary obstacle to effective ocular medication delivery lies in the eye’s anatomical barriers. These include the tear film, cornea, vitreous, and the crucial blood–aqueous and blood–retina barriers (Figure 3). While these barriers function as defense mechanisms against external harmful agents, they simultaneously pose significant challenges to the bioavailability of ocular drugs. The blood–retina barrier significantly impedes the transfer of drugs from the systemic circulation to the posterior eye segments. The other barriers primarily obstruct the absorption of externally applied drugs into both the anterior and posterior segments of the eye.

### 5.2. Anterior Segment Diseases

#### 5.2.1. Glaucoma

Glaucoma, often described as the “silent thief of sight,” insidiously advances without early noticeable symptoms, resulting in often belated detection and management. As a leading cause of irreversible blindness worldwide, glaucoma is hallmarked by glaucomatous optic neuropathy. This neuropathy is evidenced by the cupping or hollowing of the optic disc, degeneration of axons, apoptotic loss of retinal ganglion cells, and consequently, permanent vision impairment [146]. The condition’s complex origins involve an interplay of genetic and environmental factors. While intraocular pressure (IOP) is recognized as a consistent risk factor, it is not a definitive marker for all primary open-angle glaucoma (POAG) cases, indicating that elevated IOP is not a universal trait of the disease. Nonetheless, the control of IOP has retained its position as the primary therapeutic target in glaucoma management. This emphasis persists because modulation of IOP currently represents the only adjustable risk factor known to affect the disease’s progression, which underscores the critical nature of IOP management in glaucoma treatment strategies [147]. The application of biopolymer-based drug delivery systems in the management and treatment of glaucoma has been extensively researched. Among these, chitosan, hyaluronic acid, and silk fibroin are noteworthy due to their promising potential, as demonstrated in various studies.

Chitosan:

Chitosan, a biocompatible and biodegradable polymer, has shown significant potential as a nanoparticle-based polymeric drug delivery system for glaucoma treatment. Studies highlight the effectiveness of chitosan nanoparticles in enhancing drug delivery to the eye, addressing challenges such as low bioavailability and limited drug retention times associated with conventional drug delivery systems. Various studies have indicated that chitosan nanoparticles can improve the delivery and efficacy of antiglaucoma agents by overcoming barriers such as poor residence time and low corneal permeability, which are common limitations of traditional ocular pharmaceutical products [148]. Additionally, the mucoadhesive properties of chitosan also contribute to its potential applications, aiding in sustained drug release and enhanced therapeutic performance for glaucoma management [149].

For instance, Ameeduzzafar et al. studied the efficacy of carteolol-loaded chitosan nanoparticles. These nanoparticles exhibited efficacy in enhancing the ocular hypotensive effect of carteolol, as demonstrated by in vitro and ex vivo studies, which revealed effective permeation through the cornea and a prolonged reduction in intraocular pressure in betamethasone-induced glaucoma rabbits [150]. In another study, Franca et al. explored the effectiveness of chitosan and hydroxyethyl cellulose-based ocular inserts for the prolonged administration of dorzolamide, which resulted in a decrease in intraocular pressure [151]. These findings suggest that polymeric-based ocular inserts may have potential applications for sustained release of dorzolamide in the management of glaucoma.

Overall, the promising potential of chitosan as a nanoparticle-based drug delivery system for glaucoma treatment is noteworthy. Compared to traditional drug delivery systems, it has the potential to overcome certain limitations and provide sustained drug release, thereby offering a significant improvement in the management of glaucoma.

Silk Fibroin:

Silk fibroin, a biopolymer derived from silk and recognized for its versatility, has emerged as a promising material in the field of ophthalmology, particularly in the context of glaucoma treatment. Its exceptional biocompatibility, biodegradability, and mechanical properties have rendered it an ideal candidate for the development of nanoparticles designed for targeted drug delivery within the eye [152]. For instance, Zhang et al. conducted an extensive study on the use of silk fibroin film for wound healing using both animal models and a randomized controlled clinical trial. This study revealed that silk fibroin films expedite wound healing and minimize adverse effects, emphasizing their potential for ocular therapeutic applications [153].

In another study, Suzuki et al. aimed to optimize silk fibroin membranes to enhance their physical properties for use in retinal implantation. They accomplished this by incorporating poly(ethylene glycol) (PEG) and horseradish peroxidase (HRP) into fibroin membranes, resulting in improved handling and increased biocompatibility. These modified membranes were capable of supporting the construction of functional retinal pigment epithelium with minimal impact on the surrounding tissue [154].

The utilization of silk fibroin in various medical applications has opened new possibilities for its use in the treatment of glaucoma. The potential of this material for wound healing and retinal repair suggests that it may be effective as a targeted therapy for glaucoma.

Hyaluronic Acid (HA):

HA has gained significant attention as a drug delivery system for various medical conditions, including glaucoma. The growing interest in utilizing HA is due to its properties such as water solubility, biodegradability, biocompatibility, low toxicity, and non-immunogenicity within the body [155,156]. For instance, Desai et al. demonstrated the successful co-delivery of timolol and HA from contact lenses for an extended period to potentially treat glaucoma by conducting both in vitro and in vivo evaluations [157]. This study highlights the potential of HA-based drug delivery systems to provide sustained release of medications for glaucoma treatment. Egbu et al. demonstrated the potential of antibody-loaded collapsible HA hydrogels for intraocular delivery. The results suggest that poly (ethylene glycol) diacrylate-poly N-isopropylacrylamide-hyaluronic acid (PEGDA-pNIPAAM-HA) has the potential for further development as a formulation to prolong the intraocular release of proteins, which could have implications for ocular drug delivery and treatment of various conditions, including glaucoma [158].

Overall, the use of HA in drug delivery systems for glaucoma treatment holds great potential due to its biocompatibility, targeted drug delivery capabilities, and versatility in various drug delivery formulations. Its potential to enhance drug delivery efficiency, reduce toxicity, and provide targeted drug delivery makes it a valuable component for the development of effective glaucoma treatment strategies.

#### 5.2.2. Dry Eye Disease

Dry eye disease (DED), also known as keratoconjunctivitis sicca, is a complex condition marked by inadequate or suboptimal tear quality, leading to discomfort, visual disturbances, and tear film instability [159]. This condition, which often results in ocular surface inflammation and damage, is characterized by increased tear film osmolarity. The etiology of DED is diverse, encompassing factors such as aging, certain systemic conditions such as Sjogren’s syndrome, specific medications, environmental influences, lifestyle factors, and hormonal changes [160].

Therapeutic strategies for DED vary based on its severity. For milder instances, treatment may involve the use of artificial tears and ointments. In contrast, more severe cases might necessitate the application of topical corticosteroids, immunosuppressants, or autologous tear therapy. Each treatment option carries its own set of limitations and side effects. For instance, artificial tears require frequent application, necessitating patient compliance. Long-term use of topical steroids is associated with risks such as increased intraocular pressure and cataracts. Autologous tear therapy, while effective, is cost-intensive and requires multiple healthcare visits for blood draws [161].

Moreover, DED poses challenges to ocular drug delivery. It can reduce the residence time of topically applied drugs and elevate the risk of systemic absorption. Additionally, increased tear turnover associated with DED can diminish the effectiveness of topical medications. To counter these issues and maintain ocular surface health, alternative therapeutic strategies have been developed to improve drug delivery in the context of DED. Although various advancements have been made in biopolymeric drug delivery systems for DED, some of the most notable ones include the development of thermo-responsive hydrogels for enhanced drug solubility and retention, biodegradable drug delivery systems via 3D bioprinting, and long-acting mucoadhesive thermogels for sustained treatment.

Thermo-Responsive Hydrogels: Han et al. developed a novel hybrid thermo-responsive hydrogel. This formulation, which incorporates mono-functional polyhedral oligomeric silsesquioxane, polyethylene glycol, and polypropylene glycol, has been shown to improve the water solubility of FK506, an immunosuppressive drug, while also extending its retention time on the ocular surface. In a murine model of dry eye, this hydrogel demonstrated superior effectiveness compared to traditional FK506 treatments, making it a potentially effective drug delivery system for DED [162].

Biodegradable Drug Delivery Systems: Park et al. developed a novel lens-type biodegradable drug delivery system using 3D bioprinting technology, which incorporates gelatin methacryloyl, hyaluronic acid, antibiotics, and conjunctival epithelial cells. The system is engineered for a controlled degradation rate, which could potentially facilitate the treatment and regeneration of the epithelium in patients with DED, thus offering a promising approach to ocular drug delivery [163].

Long-Acting Mucoadhesive Thermogels: Luo et al. developed a long-acting, mucoadhesive thermogel, representing another advancement in biopolymeric drug delivery for DED. This system uses a combination of gelatin, poly(N-isopropylacrylamide), and Helix pomatia agglutinin, facilitating the prolonged and effective delivery of epigallocatechin gallate. In a rabbit model of DED, the thermogel was found to effectively repair the corneal epithelium and alleviate symptoms for an extended period, indicating its potential utility in the long-term management of DED [164].

Drug-loaded Nanocarrier: Yang et al. developed an innovative cerium oxide-based nano-system for delivering gabapentin (GBT) to treat DED. This nanosystem is modified with biopolymers, specifically thiolated gelatin, and cross-linked with glutaraldehyde. These modifications enhanced its biocompatibility, increased cellular uptake, and improved mucin-binding efficiency, leading to prolonged ocular retention time and better therapeutic effect. Results also demonstrated multiple potential therapeutic uses, such as antioxidant, anti-inflammatory, antiangiogenic, antiapoptotic, and neuroprotective effects. In a rabbit model of DE, the GBT-loaded nanoceria showed superior therapeutic effects compared to GBT without the nanocarrier, including corneal damage repair and tear secretion stimulation. Thus, it is a promising candidate for DE treatment [159].

Overall, various studies of biopolymeric drug delivery systems in the context of DED have revealed the potential to improve therapeutic efficacy. Research in this area has emphasized the versatility and potential of biopolymers in addressing the limitations of conventional DED treatments. The application of these novel biopolymeric systems could be beneficial for DED treatment and management.

#### 5.2.3. Cataracts

Cataracts are characterized by the clouding or opacification of the normally clear lens of the eye, which impedes the passage of light to the retina, leading to vision impairment. This condition is predominantly seen in older individuals but can affect all age groups, including infants. Cataracts may vary in severity and can be bilateral. In the early stages, the disease process may not significantly impact daily activities. However, as it progresses, particularly after the fourth or fifth decade, the lens can become completely opaque, substantially interfering with routine activities [165].

Phacoemulsification is the gold standard for cataract surgery, with a high success rate, though it is not without risks. Immediate postoperative complications can include minor discomfort and elevated intraocular pressure; short-term complications may involve cystoid macular edema and posterior capsular opacification (PCO), treatable with YAG laser capsulotomy. Less common but serious complications, such as posterior capsule rupture, endophthalmitis, as well as retinal detachment, require prompt management. Given the potential complications of cataract surgery and the high prevalence of cataracts, research is increasingly focused on preventative strategies to obviate the need for surgical intervention. This shift reflects a growing interest in addressing the condition proactively, exploring pharmacological approaches that may delay or prevent the onset of cataracts [146]. In regard to cataracts, thermoresponsive multi-drug delivery systems, the development of intraocular and external ophthalmic implants and polymers, and liposomal drug delivery systems are particularly noteworthy because of their innovative nature and potential in addressing the limitations associated with conventional approaches.

Intraocular and External Ophthalmic Implants and Biopolymers: The utilization of novel intraocular and external ophthalmic implants, polymers, and nanotechnology is becoming increasingly important for enhancing postoperative drug delivery in cataract surgery [166]. Recent advancements in this area include the application of supercritical CO_2_ to load gatifloxacin into hydrophobic foldable intraocular lenses, offering a solvent-free approach to reduce the risk of postoperative endophthalmitis [167]. Additionally, the development of liquid-like layer-coated intraocular lenses aims to prevent posterior capsular opacification and reduce intraocular inflammation [168]. Furthermore, the introduction of a polydopamine-based photodynamic coating on intraocular lenses is effective in preventing PCO by eliminating lens epithelial cells [169]. Finally, the integration of an antibacterial nanopillar array on intraocular lenses represents a novel approach to capture and eradicate bacteria, thereby reducing the risk of postoperative endophthalmitis [170]. These recent advancements collectively represent a shift towards safer and more effective post-cataract surgery outcomes.

Thermoresponsive Multi-Drug Delivery Systems: Recent research has focused on the development of thermoresponsive multidrug delivery systems for post-cataract surgery care, aiming to minimize complications such as inflammation, infection, and posterior capsular opacification (PCO). For instance, Yan et al. introduced a temperature-sensitive hydrogel system for the intracameral delivery of dexamethasone, moxifloxacin, and genistein, demonstrating its efficacy in reducing cell proliferation and migration in vitro [171]. Similarly, Zhang et al. developed a thermoresponsive nanocomposite combining prednisolone acetate and levofloxacin, which showed reduced ocular inflammation and enhanced antibacterial activity with minimal cytotoxicity [172]. Cheng et al. also contributed to this field by developing a chitosan-based hydrogel for dual drug delivery, which effectively managed postoperative inflammation and bacterial infection in an ex vivo rabbit model [173]. Collectively, these studies demonstrate the potential of thermoresponsive drug delivery systems for enhancing post-surgical outcomes and pave the way for future innovations in postoperative drug delivery.

Liposomal Drug Delivery Systems: A major research area in cataract treatment involves exploring the enhanced bioavailability and circulation time of Coenzyme Q10 (CoQ10), a potential anti-cataract agent. In a study, Yang focused on the development of PEG-liposomes for this purpose. These liposomes were engineered with varying molar ratios of PEG-lipids and cholesterol to optimize delivery. The main parameters, including morphology, particle size, zeta potential, and in vivo pharmacokinetics, were assessed [174]. The study employed the protamine aggregation method to ascertain the encapsulation efficiency of the optimized formulation, determined through a Box–Behnken design. A significant finding was the positive correlation of liposome particle size with cholesterol content and a negative correlation with PEG-lipid content. In vivo analyses demonstrated a significant enhancement in the half-life and bioavailability of CoQ10, implying potential improvements in preventive efficacy for cataract formation [174]. In another study, Huang et al. focused on the development of chitosan-coated liposomes containing lanosterol and hesperetin. The preparation involved thin film evaporation and active extrusion, with comprehensive characterization techniques employed to assess the formulation. The in vivo results were promising, showing an effective delay in cataract progression. The liposomes exhibited stability for over 60 days and were found to be non-toxic. Additionally, they enhanced the body’s antioxidant defense system [175]. This study demonstrates the potential of these liposomes as a viable, safe, and non-surgical option for cataract prevention.

### 5.3. Posterior Segment Diseases

Posterior eye diseases, encompassing conditions affecting the vitreous, retina, and choroid, present significant challenges in ophthalmic care due to the complex eye structure obstructing efficient penetration, absorption, and residence time for medications. Conventional treatments often fall short in addressing the complexities of these diseases. Conditions such as age-related macular degeneration (AMD), diabetic retinopathy (DR), and uveitis are the focal points of posterior segment disease research. Current treatments, including intravitreal injections and implants, face drawbacks including limited drug retention, frequent administration, ocular irritation, and potential complications such as retinal detachment and increased intraocular pressure (IOP). The need for sustained drug release and targeted delivery to the posterior segment necessitates novel approaches to enhance treatment efficacy. Biopolymers, characterized by biodegradability, biocompatibility, non-toxicity, tunable degradation rates, and high drug loading ratios, have shown promise for controlled delivery of therapeutic agents, providing sustained and localized delivery to targeted regions [176]. This section of the review explores the limitations of existing therapies, delves into the potential of bio-polymer-based drug delivery systems, and outlines current challenges that demand further research in the field of posterior eye disease management [176,177].

Another challenge for efficient posterior segment drug delivery is mucin protein and mucus acting as barrier and retention sites, effecting drug absorption and penetration. Chitosan, a cationic polymer, has shown superior mucoadhesive and mucopenetrating properties due to its positive charges interacting with the negatively charged residues of mucin protein, enhancing residence time. Other biopolymers have also been investigated for their mucoadhesive properties. For example, alginate can swell and interpenetrate its chains with mucins protein on the ocular surface [178]. A study demonstrated that both cationic nanosuspension and drug-core mucus-penetrating particles are effective anterior tissue drug delivery carriers with cationic nanosuspension achieved a higher bioavailability [179]. To efficient ocular drug delivery to the posterior segment, Yanyan’s team designed a hybrid nanocomposite based on layered double hydroxide and carboxymethyl chitosan derivatives for targeted delivery of dexamethasone disodium phosphate. The nanocomposite demonstrated non-cytotoxicity and enhanced permeability in vitro. In vivo studies revealed a substantial increase in precorneal retention, offering a potential strategy for non-invasive, topical administered drug delivery to the posterior eye segment [180]. 

#### Age-Related Macular Degeneration (AMD) & Diabetic Retinopathy (DR)

AMD and DR stand out as prominent posterior eye diseases, driving numerous studies to innovate new drug delivery systems. Conventional treatments for these conditions often involve anti-vascular endothelial growth factor (anti-VEFG) medications such as bevacizumab, ranibizumab, and aflibercept. However, traditional drug delivery methods pose challenges related to patient compliance, requiring frequent drug injections and invasive drug delivery methods that might lead to potential side effects, burst drug release, non-sustained drug release, and more. Jiang et al. synthesized biodegradable polymer-based core–shell microparticles to address these challenges. The chitosan–polycaprolactone core–shell microparticles were designed with electrostatic and physical interactions to control protein diffusion, increase drug loading capacity, reduce uncontrollable burst release, and extend the period of drug release. Another study tackled the issue of short-period drug release by developing a PCL–chitosan-based bi-layered capsule with a central hollow design for higher drug loading and long-term drug release to be extended successfully to over one year [181]. Other chitosan-based polymeric nanocarrier, chitosan grafted-poly(ethylene glycol) methacrylate nanoparticles, were developed via a double linking process [182]. 

Llabot et al. delve into the in vitro characterization of human serum albumin nanoparticles stabilized with Gantrez^®^ES-425. These nanoparticles, loaded with antiangiogenic drugs such as seramin and bevacizumab, demonstrated antiangiogenic effects and reduced levels of inflammation and fibrosis compared to free bevacizumab. Together, these studies showcase the potential of albumin nanoparticles as a topical treatment for corneal neovascularization (CNV) and other posterior eye diseases [183,184,185]. The abundance, biocompatibility, and availability of cellulose and its derivatives offer significant advantages for this biomaterial [186]. For instance, Bessone et al. investigated ethylcellulose nanoparticles as a drug delivery system for ocular neurodegenerative disease. Their study demonstrated that melatonin-loaded ethylcellulose nanocapsules can provide a controlled topical drug release (slow-release profile) and higher corneal penetration compared to melatonin solution [187]. Furthermore, another study presented the significant effect of integrating cellulose nanocrystals (CNC) with drug-loaded, thermos-responsive poloxamer copolymer (PM). The addition of CNC reduced the critical concentration of gelation, enhancing gel strength via hydrogen bonding [188]. Yang et al. developed a carboxymethyl cellulose-based thermos-sensitive hydrogel for CNV therapy. In vivo and animal experiment results demonstrated the high transparency, biocompatibility, and sustained drug release profile for the gel. This state-of-the-art hydrogel also displayed effective inhibition of neovascularization, indicating its potential for treatment of neovascular ocular diseases, including AMD and RD [189]. In addition, increasing clinical studies explore KSI-301, an intravitreal anti-VEGF antibody biopolymer (phosphorylcholine polymer) conjugate for retinal diseases such as AMD, diabetic macular edema (DME) and retinal vein occlusion (RVO). Preclinical and early clinical trials showed promising results in KSI-301 efficacy, safety, durability, and stability. The covalently bound biopolymer offers a higher and optimized molecular size to increase intraocular durability and extends the drug-releasing period to 4–6 months [190]. These studies collectively showcase advancements in biopolymer-based drug delivery systems for posterior eye diseases, emphasizing improved efficacy, controlled drug release, and enhanced therapeutic outcomes.

Table 4 summarizes recent advancements in biopolymer-based drug delivery systems for posterior ocular diseases.

### 5.4. Uveitis

The versatile applications of biopolymer drug delivery systems extend to the treatment challenges of uveitis and neuro-ophthalmologic conditions, describing the intraocular inflammatory conditions affecting the uvea and conditions affecting the optic nerve, respectively. The uvea is the middle layer of the eye, comprising the iris, ciliary body, and choroid. The inflammation associated with uveitis can affect any or all of these components. The diverse location of the inflammatory region brings unique difficulties in targeting and efficient treatments. Current treatments for uveitis often rely on immunosuppressive and anti-inflammatory medications. However, limitations exist in the current route of drug delivery methods. Topical and oral administration and eyedrops are common, non-invasive drug delivery methods. However, they face the drawbacks of not reaching the posterior segment effectively and precisely, systemic administration with the risk of side effects, and necessitating frequent and increased administration. Intravitreal injections and drug-eluting implants have been introduced to improve drug delivery efficiency. However, they are more invasive procedures, and repeated injections are associated with increased risks of ocular hypertension, ocular irritation, patient discomfort, and other complications [201]. In this subsection, we explore the advancement in the use of biopolymers in drug delivery systems and how the nature of biopolymers offers the potential for sustained drug release, precise targeting, and reduced side effects in the context of uveitis.

#### 5.4.1. Applications of Biopolymer-Based Hydrogels in Drug Delivery

For instance, to address treatment difficulties in bacterial endophthalmitis, Bao et al. engineered a glycol chitosan crosslinked oxidized hyaluronic acid hydrogel film, showcasing promise for the stepwise delivery of dexamethasone (Dex), followed by sustained levofloxacin (Lev) release. Notably, the oxidation degrees of oxidized hyaluronic acid (OHA) influenced the swelling ratio of the hydrogel films. In vitro studies have presented the potent anti-bacterial effects of the engineered hydrogel film against various strains and its anti-inflammatory activity by down-regulating inflammatory cytokines [191]. Other studies also investigated postoperative endophthalmitis treatments using biopolymer-based hydrogels. In vitro studies reveal a sustained-release profile of Lev-loaded chitosan-based hydrogel, indicating its long-term anti-bacterial properties [192]. Cheng et al. later engineered a chitosan–gelatin-based hydrogel loaded with prednisolone acetate (PA) and Lev-loaded nanoparticles. The fabricated hydrogel demonstrates sustained release over 7 days and, importantly, showcases anti-inflammatory and anti-bacterial properties in both in vitro TNF-α-damaged corneal epithelial cells and an ex-vivo rabbit model of S. aureus keratitis [173]. Continuing the advancements, Khan’s team designed and optimized a water-insoluble corticosteroid triamcinolone acetonide (TAA)-loaded in situ gel, using reacted tamarind seed xyloglucan (RXG) polysaccharide and carrageenan (CRG) polysaccharide polymers. The optimized TAA-loaded dual responsive in situ gel exhibited favorable rheological properties, transitioning from a flowable state at 25 °C to a robust gel at 35 °C, with strong mucoadhesion that prolongs retention time, toleration, and spreadability on the cornea. In comparison with TAA suspension, the optimized in situ gel improved the overall drug exposure and maintained a high drug concentration and retention time in the vitreous humor [202]. Moreover, in the pursuit of non-invasive drug delivery for efficient uveitis treatment, hydrogel eye drops emerge as a promising carrier for adalimumab (ADA) and diclofenac sodium (DIC). In vitro experiments revealed that the low-deacetylated chitosan and β-glycerophosphate hydrogels are biocompatible with both human corneal and retinal pigment epithelial cells. The rabbit in vivo pharmacokinetics study demonstrated that ADA-loaded hydrogel eye drops have a superior permeation rate and retention time compared to free ADA [203].

#### 5.4.2. Biodegradable Nano-Based Drug Delivery Systems

The cutting-edge frontier of biodegradable nano-based drug delivery systems further refines drug delivery efficacy and precision, which are crucial in ocular inflammatory disease management to avoid potential side effects and optimize therapeutic outcomes. For instance, chitosan–tripolyphosphate nanoparticles emerge as a promising strategy for enhancing the inhibitory effect of natamycin due to their high drug loading capacity, which combats the poor water solubility drawback of natamycin. The paper demonstrated the slow drug release and antifungal properties of natamycin-loaded nanoparticles in vitro, shadowing their potential application in fungal keratitis treatment [204]. Continuing the exploration, two studies showcase the potential of biodegradable poly(lactic-co-glycolic acid) (PLGA) nanoparticles for controlled drug release in treating noninfectious uveitis. Firstly, biodegradable carboxyl-terminated PLGA nanoparticles are examined for the controlled release of dexamethasone sodium phosphate (DSP) via subconjunctival injection. The encapsulation of DSP, PLGA, and nanoparticles was achieved through a zinc ionic “bridge.” In ex vivo and in vivo experimental autoimmune uveitis (EAU) rat models, DSP-NP displayed sustained drug release and efficacy in reducing inflammation [193]. Secondly, methoxy poly (ethylene glycol)-poly(dl-lactide-co-glycolic acid) (mPEG-PLGA) nanoparticles loaded with triamcinolone acetonide (TA) demonstrate sustained release over 45 days, providing enhanced anti-inflammatory effects compared to TA alone. In addition, intravitreal-injected TA-loaded mPEG-PLGA nanoparticles demonstrated good biocompatibility and controlled release in rat models [194]. Moreover, a separate investigation explored the potential of chitosan-coated PLGA nanoparticles serving as carriers for atorvastatin calcium, which were further incorporated into thermosensitive gels. This research extensively assessed various parameters, including size (ranging from 80.0 to 190.0 ± 21.6 nm), gelation characteristics, viscosity, in vitro release patterns, and anti-inflammatory efficacy. This innovative approach not only addresses the challenges associated with the limited oral bioavailability of atorvastatin calcium but also establishes a foundation for site-specific actions within ocular tissues [195].

Additionally, biodegradable biopolymer-based ocular drug-eluting implants, such as Ozurdec^®^ and Dexycu^®^, could be utilized to treat uveitis or postsurgical inflammation safely [196,197]. Ozurdec is a single-use intravitreal implant containing dexamethasone in the NOVADUR, a PLGA-based polymer drug delivery system. The PLGA backbone is capable of slow degradation into lactic acid and glycolic acids in the vitreous, reducing the need for frequent drug administration. Recent clinical testimonies of the commercially available, FDA-approved intravitreal implant Ozurdec demonstrated its effectiveness in delivering localized dexamethasone and providing visual acuity without causing adverse side effects. However, more research is required to lower the risks of causing increased intraocular pressure and cataract formation [198,199,200].

Overall, a biopolymer-based drug delivery system holds the potential to overcome the drawbacks associated with current treatments. These systems can improve drug stability, prolong therapeutic effects, and reduce the need for frequent invasive interventions. While biopolymer-based DDS shows great promise, several challenges persist. Achieving optimal drug release kinetics, ensuring precise targeting, and addressing potential immunogenicity issues are areas that require further exploration. In addition, the development of personalized formulations and the optimization of biopolymer characteristics demand future research efforts.

## 6. Current Challenges and Future Directions

In the field of ophthalmology, the application of biopolymers in oculoplastic and orbital surgery presents various challenges that need to be addressed. One of the primary concerns is the design and selection of implants, where challenges such as implant migration, postoperative infections, inadequate prosthesis motility, and the need for multiple surgeries are frequently encountered [47,205,206].

In terms of hydrogels, challenges include achieving the desired degradation period, mechanical properties, and cell affinity for specific ophthalmic applications [106,207]. Additionally, assessing drug penetration and retention in hydrogel systems for ocular drug delivery remains a challenge [208].

Additionally, the current state of research primarily focuses on the development of early-stage prototypes and in vitro settings. However, the transition from these developments to clinical applications presents significant challenges. This process involves progressing to in vivo and preclinical studies, addressing regulatory requirements, and considering patient-specific factors such as ocular health status, allergies, and sensitivity.

Looking towards the future, there is significant potential for advancements in biopolymer applications in ophthalmology, with a focus on developing novel materials with enhanced functionalities, such as improved antibacterial properties, angiogenesis, and tailored drug release kinetics. These developments are essential for addressing postoperative complications, optimizing therapeutic efficacy, and overcoming ocular barriers to increase drug bioavailability. Future research should prioritize the improvement of bicompatibility, reduction of tissue reactions, and comprehensive optimization of biopolymer characteristics, including biodegradability, crosslinking capabilities, and mechanical properties. The integration of emerging technologies, such as nanotechnology and smart materials, into biopolymer-based drug delivery systems, along with features such as thermosensitivity and redox-responsive behavior, opens new avenues for personalized medicine.

## 7. Conclusions

In this review, we have explored the extensive range of applications for biopolymers in ophthalmic medicine, particularly in the development of intraocular lenses, various prosthetics, and novel drug delivery systems. Our findings have not only enhanced the understanding of current techniques but have also paved the way for future advancements in the field, focusing on improving patient comfort and considering environmental factors.

The exploration of oculoplastic and orbital surgery devices, applications of hydrogels in ophthalmology, and biopolymeric drug delivery systems has laid the foundation for future research and clinical applications. We have provided a platform for evaluating the efficacy and safety of biopolymers, concentrating on their physical properties, compatibility with biological tissues, and clinical performance in various areas of ophthalmology.

The collaboration between material scientists and ophthalmologists, as advocated in this article, signifies a paradigm shift towards a more innovative and patient-centered approach to ophthalmic medicine. This collaboration will continue to drive the field toward realizing the full potential of biopolymers in enhancing the quality of life of patients by revolutionizing ophthalmic device technology and overcoming the current challenges in the field.

## Figures and Tables

**Figure 1 polymers-16-01717-f001:**
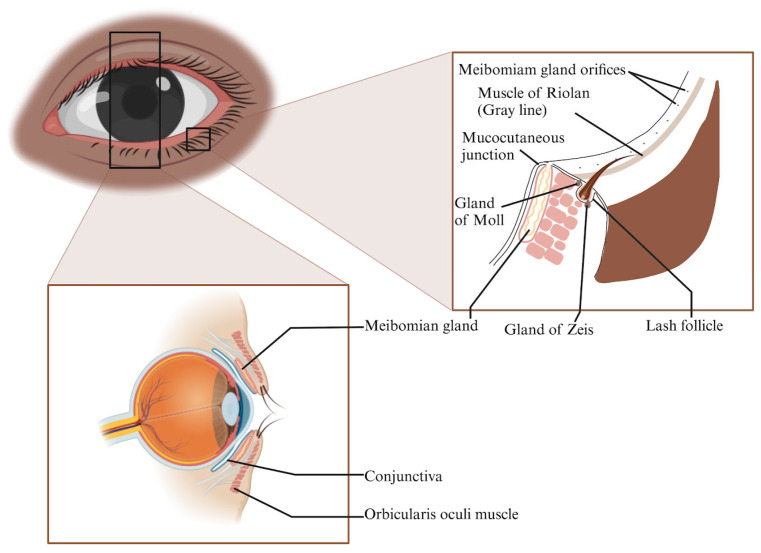
Illustration of marginal eyelid structures and associated glands. The insets show the meibomian gland orifices along the lid margin, Riolan’s muscle, mucocutaneous junction, and accessory glands of Zeis and Moll. Permission to reprint granted by Wu et al. (2024), from the publication titled Polymers and Biomaterials for Posterior Lamella of the Eyelid and the Lacrimal System [68].

**Figure 2 polymers-16-01717-f002:**
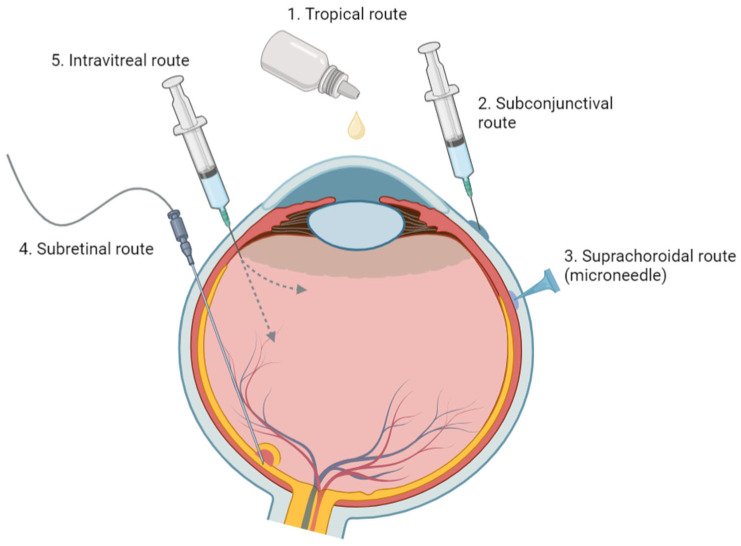
Overview of various ophthalmic medication delivery routes. This figure illustrates the spectrum of administration methods utilized in ophthalmic medicine, encompassing topical, subconjunctival, intravitreal, suprachoroidal, and subretinal techniques.

**Figure 3 polymers-16-01717-f003:**
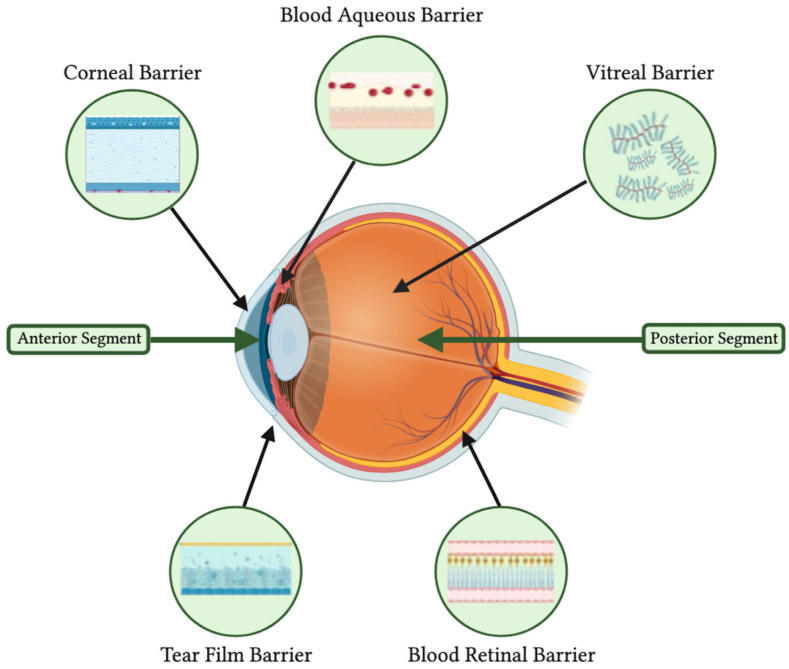
Overview of anatomical and physiological barriers to pharmacologic intervention in ophthalmic administration. This diagram outlines the anatomical and physiological barriers encountered in the delivery of ophthalmic medications, specifically highlighting the corneal barrier, blood–aqueous barrier, vitreal barrier, tear film barrier, and blood–retinal barrier.

**Table 3 polymers-16-01717-t003:** Summary of recent applications of biopolymer-based hydrogel in corneal repair.

Polymer	Advantages	Disadvantages/Limitation	Applications	References
Gelatin methacrylate (GelMA)	Excellent biocompatibility, good mechanical stability, cell adhesive properties, MMP-degradable	May require modification for enhanced properties, cost	- Corneal stromal regeneration - Sutureless keratoplasty	[128,129,130,131,132]
Collagen-based hydrogels	Natural component of the cornea, high biocompatibility, mimics corneal structure	Limited mechanical strength, potential for immunogenicity	- Corneal tissue engineering - Artificial corneal substitute	[134,135,137]
Silk fibrin	Biocompatibility, rapid gelation, tunable degradability, transparency	Potential variability in degradation rate, source-dependent	- Corneal stromal regeneration	[139]
Peptide-based, Poly-ε-lysine (pεK)	Mimics native ECM, porosity, transparency, swell-capability	Requires cross-linking for desired properties	- Substrate for corneal endothelial cell growth	[140,141,142]
Nanocellulose	Abundant, Biocompatible, High mechanical strength, Sustainable	Requires chemical modification for enhanced properties, Potential issues with long-term stability	- Reinforcement in collagen hydrogels - Drug delivery	[137,138]
Mucin	Abundant, biocompatible, natural component in the eye	Lack of transparency in some formulations, not widely studied for ocular use	- Potential for corneal regeneration - Drug delivery	[143,144,145]

**Table 4 polymers-16-01717-t004:** Recent advancements in biopolymer-based drug delivery system for posterior ocular diseases.

Polymer	Advantages/Advancements	Disadvantages/Limitations	Applications	References
Chitosan	- Superior mucoadhesive and mucopenetrating properties - Controlled protein diffusion, high drug loading, reduced burst release - Long-term anti-bacterial and anti-inflammatory properties	- Potential cytotoxicity at high concentrations - Some formulations require optimization for human use	- Drug delivery for posterior segment diseases - Sustained drug delivery for AMD, DR and uveitis	[143,180,181,182,191,192]
Human serum albumin nanoparticles	- Antiangiogenic effects, reduced inflammation and fibrosis	- Stability and storage issues	- Topical treatment for corneal neovascularization (CNV)	[183,184,185]
Cellulose and its derivatives	- Controlled drug release, high corneal penetration - Enhanced gel strength, reduced gelation concentration	- Potential toxicity - Some formulations require further testing	- Drug delivery for ocular neurodegenerative disease - Thermo-responsive hydrogel for drug delivery	[187,188,189]
Phosphorylcholine polymer	- Extended drug release (4–6 months), high intraocular durability	- Clinical trials still ongoing	- Treatment for AMD, DME, RVO	[190]
Glycol chitosan-oxidized hyaluronic acid hydrogel	- Stepwise drug release, anti-inflammatory, anti-bacterial	- Swelling ratio dependent on oxidation degree	- Treatment for bacterial endophthalmitis	[191]
Poly(lactic-co-glycolic acid) (PLGA) nanoparticles	- Sustained drug release, biocompatibility	- Potential for immune response	- Treatment for non-infectious uveitis	[193,194,195]
PLGA-based drug-eluting implants (ozurdex, dexycu)	- Prolonged drug release, localized delivery, FDA approved	- Risk of increased intraocular pressure and cataract formation	- Drug-eluting implants for uveitis, postsurgical inflammation	[196,197,198,199,200]

## Data Availability

Not applicable (no new data were created).

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
