# Peer review of "Biopolymeric Innovations in Ophthalmic Surgery: Enhancing Devices and Drug Delivery Systems"

_polymers, 2024, doi:10.3390/polym16121717_

Round 1
Reviewer 1 Report
Comments and Suggestions for Authors
There are some points that authors are required to revise
1. The review has comprehensive information where some information is different from another, therefore, a suggestion for authors to use atleast 2 or more tables so that the readability of the paper can be improved.
2. The author's choice of title is quite broad. There are various biopolymer formulations and drug delivery cases/information, which were left out in the current manuscript, therefore choose an appropriate title for the paper.
3. Eye cornea materials: There is vast literature available for eye cornea materials and one of the hot topics in ophthalmology, where most commercial-grade polymers are synthetic versions. However, the author atleast needs to compile a table where polymer advantages, disadvantages, and limitations can be compared, this would be more appealing to the readers.
4. No information on cellulose-based polymers or derived polymers was considered in the current manuscript, seeing cellulose being the most abundant biopolymer, therefore must be included in the manuscript.
Author Response
Dear Reviewer,
We sincerely appreciate your thorough review and valuable comments on our manuscript. We have carefully addressed each of your suggestions and made the following revisions:
Tables for Improved Readability: We have included two additional tables in our updated manuscript to enhance readability and provide a clearer presentation of the comprehensive information. These tables help to organize the data more effectively.
Title Adjustment: We agree that the original title was broad. We have revised the title to better reflect the specific content and focus of our manuscript. The new title now reads “Biopolymeric Innovations in Ophthalmic Surgery: Enhancing Devices and Drug Delivery Systems”, which accurately captures the scope of the biopolymer applications and drug delivery systems discussed.
Eye Cornea Materials: A new table has been added that compares the advantages, disadvantages, and limitations of various commercial-grade synthetic polymers used for eye cornea materials. This table aims to provide a comprehensive overview for our readers and highlight the key differences among these materials.
Cellulose-Based Polymers: We have included a section on cellulose-based polymers and their derivatives, acknowledging their abundance and significance as biopolymers. This addition ensures that our manuscript covers a broader spectrum of biopolymers relevant to ophthalmology.
Thank you once again for your insightful feedback. We believe these revisions significantly improve the quality and clarity of our manuscript.
Best regards,
Reviewer 2 Report
Comments and Suggestions for Authors
This is a nice review about biopolymers in ophthalmology. I recommend it for publication after the following minor points are addressed.
1. The authors should discuss more about mucin in this review since mucin is a biopolymer existed in eyes.
2. It is better to add one table or one figure to summarize the current studies in this topic.
3. Line 788-790, one recent study (ACS Nano 2023, 17, 24, 25118–25135) should be included to support such a claim.
Author Response
Dear Reviewer,
Thank you for your positive feedback and for recommending our manuscript for publication with minor revisions. We have addressed each of your comments as follows:
Discussion on Mucin: We have expanded the discussion on mucin, emphasizing its role as a biopolymer in the eyes. This addition provides a more comprehensive overview of biopolymers relevant to ophthalmology.
Summary Table/Figure: To summarize the current studies in this topic, we have added a new table that consolidates and highlights the key findings of recent research. This table aims to provide readers with a quick reference to the state of the art in this field.
Inclusion of Recent Study (ACS Nano 2023, 17, 24, 25118–25135): We have included the mentioned study in our manuscript to support the claim made in lines 788-790. This addition strengthens the argument and provides readers with the latest relevant research.
We appreciate your constructive comments and believe that these revisions have enhanced the overall quality of our manuscript.
Best regards,